# A 62-ka geomagnetic palaeointensity record from the Taymyr Peninsula, Russian Arctic

Stephanie Scheidt[1], Matthias Lenz[1], Ramon Egli[2], Dominik Brill[3], Martin Klug[4], Karl Fabian[4,5], Marlene M. Lenz[1], Raphael Gromig[1], Janet Rethemeyer[1], Bernd Wagner[1], Grigory Federov[6,7], Martin Melles[1]

[1]Institute of Geology and Mineralogy, University of Cologne, Cologne, 50674, Germany

[2]Central Institute for Meteorology and Geo-dynamics (ZAMG), Vienna, 1190, Austria

[3]Institute of Geography, University of Cologne, Cologne, 50674, Germany

[4]Geological Survey of Norway (NGU), Trondheim, 7491, Norway

[5]Department of Geoscience and Petroleum, Norwegian University of Science and Technology, Trondheim, 7491, Norway

[6]Institute of Earth Sciences, St. Petersburg State University, St. Petersburg, 199034, Russia

[7]Arctic and Antarctic Research Institute, St. Petersburg, 199397, Russia

*Correspondence to*: Stephanie Scheidt (Stephanie.Scheidt@uni-koeln.de)

**Short summary**

Levinson-Lessing Lake in northern central Siberia provides an exceptional opportunity to study the evolution of the Earth's magnetic field in the Arctic. This is the first study carried out at the lake that focus on the palaeomagnetic record. It presents the relative palaeointensity and palaeosecular variation of the upper 38 m of sediment core Co1401, spanning ~62 ka. A comparable high-resolution record of this time does not exist in the Eurasian Arctic.

**Abstract.** This work presents unprecedented, high-resolution palaeomagnetic data from the sedimentary record of Lake Levinson-Lessing, the deepest lake in northern Central Siberia. Palaeomagnetic analyses were carried out on 730 discrete samples from the upper 38 m of the 46-m-long core Co1401, which was recovered from the central part of the lake. Alternating field demagnetization experiments were carried out to obtain the characteristic remanent demagnetization. The relative palaeointensity is determined using the magnetic susceptibility, the anhysteretic remanent magnetization, and the isothermal remanent magnetization for normalization of the partial natural remanent magnetization. The chronology of Co1401 derives from correlation of the relative palaeointensity of 642 discrete samples with the GLOPIS-75 reference curve, accelerated mass spectrometer radiocarbon ages, and optically stimulated luminescence dating. This study focuses on the part >10 ka but also presents preliminary results for the younger part of the core. The record includes the geomagnetic excursions Laschamps and Mono Lake, and resolves sufficient geomagnetic features to establish a chronology that continuously covers the last ~62 ka. The results reveal continuous sedimentation at high rates between 45 and 95 cm ka$^{-1}$. The low variability of the magnetic record compared to data sets of reference records with lower sedimentation rates may be due to a smoothing effect associated with the lock-in depths. Because Co1401 was cored without core segment overlap the horizontal component of the characteristic remanent magnetization can only be used with caution. Nevertheless, the magnetic record of Co1401 is exceptional as it is the only high-resolution record of relative palaeointensity and palaeosecular variations from the Arctic tangent cylinder going back to ~62 ka.

# 1 Introduction

The study of past variations of the Earth's magnetic field (EMF) in the Arctic is important for palaeomagnetic, palaeoenvironmental, and palaeoclimatic research. From the palaeomagnetic point of view, most of the Arctic is located inside the tangent cylinder of the inner core and thus records magnetic field variations generated by convection in the northern magneto-hydrodynamically closed compartment of the inner core. These variations are suggested to be at least partially independent from the global pattern of secular variation (St-Onge and Stoner, 2011; Lund et al., 2016). Therefore, analyses of multiple high-latitude Arctic records are needed to understand this magnetohydrodynamic regime. From the perspective of palaeoenvironmental and palaeoclimatic research, the Arctic region has a considerable impact on the climate of temperate latitudes through a variety of interactions (Christensen et al., 2013). Thus, the much higher warming recently detected in this region makes the Arctic a key region for the understanding of the Earth's climate system. For these studies, palaeomagnetic data provide a chronological tool, which is of particular importance at high latitudes, where the establishment of robust chronologies for sedimentary successions is very challenging (Alexanderson et al., 2014). Lacustrine sediment successions are particularly valuable for studying the magnetic and environmental history in the Arctic, because they often exhibit more continuous and undisturbed deposition with high accumulation rates compared to marine sediments. However, most lake sediment records in the Arctic have a restricted age range, due to glacial erosion during the Last Glacial Maximum (LGM: 26.5 ka to 19-20 ka; Clark et al., 2009) and the limited age of sedimentary basins in permafrost landscapes with small tectonic activity. Consequently, continuous sedimentary archives extending beyond the LGM are exceptionally rare in this region.

The Russian-German project PLOT (Paleolimnological Transect) aims at expanding the data base of sedimentary records in the Arctic. For this purpose, sediment cores were obtained from lakes located along a 6000 km long latitudinal transect across northern Russia. Lake Levinson-Lessing is located on the Taymyr Peninsula, 13 km northwest of lake Taymyr (Fig. 1a). The well-stratified sediment infill of this lake is a target for palaeoenvironmental and paleoclimatic studies since the 1990's. Analyses of the 22.4-m-long sediment core PG1228 revealed continuous sedimentation with nearly constant sedimentation rates of ~70 cm ka$^{-1}$ within the past ~32 ka (Ebel et al., 1999; Andreev et al., 2003). Seismic surveys suggesting the presence of up to 115 m of undisturbed sediments (Niessen et al., 1999; Lebas et al., 2019) motivated a second coring campaign in the central part of the lake basin in 2017, which yielded a 46-m-long record at site Co1401 (Fig. 1b). Scheidt et al. (2021a) show that only the upper 38 m of the associated composite core Co1401 are undisturbed and highly suitable for palaeomagnetic analyses.

In this paper, we report on the relative palaeointensity (RPI) and palaeosecular variations (PSV) of discrete samples taken from sediment core Co1401 of Lake Levinson-Lessing. Age control is gained by accelerated mass spectrometer (AMS) radiocarbon ($^{14}$C) ages, optically stimulated luminescence (OSL) dating, and correlation of the record to the most recent version of the GLOPIS-75 palaeointensity stack (Laj et al., 2014; Laj and Kissel, 2015). Since regional records of the RPI and PSV have not been available so far, this work provides new information on the behaviour of the Earth's magnetic field in the Eurasian Arctic during the Late Quaternary. The age-depth model presented also forms the chronostratigraphic backbone for palaeoenvironmental studies in the region (Lenz et al. 2022).

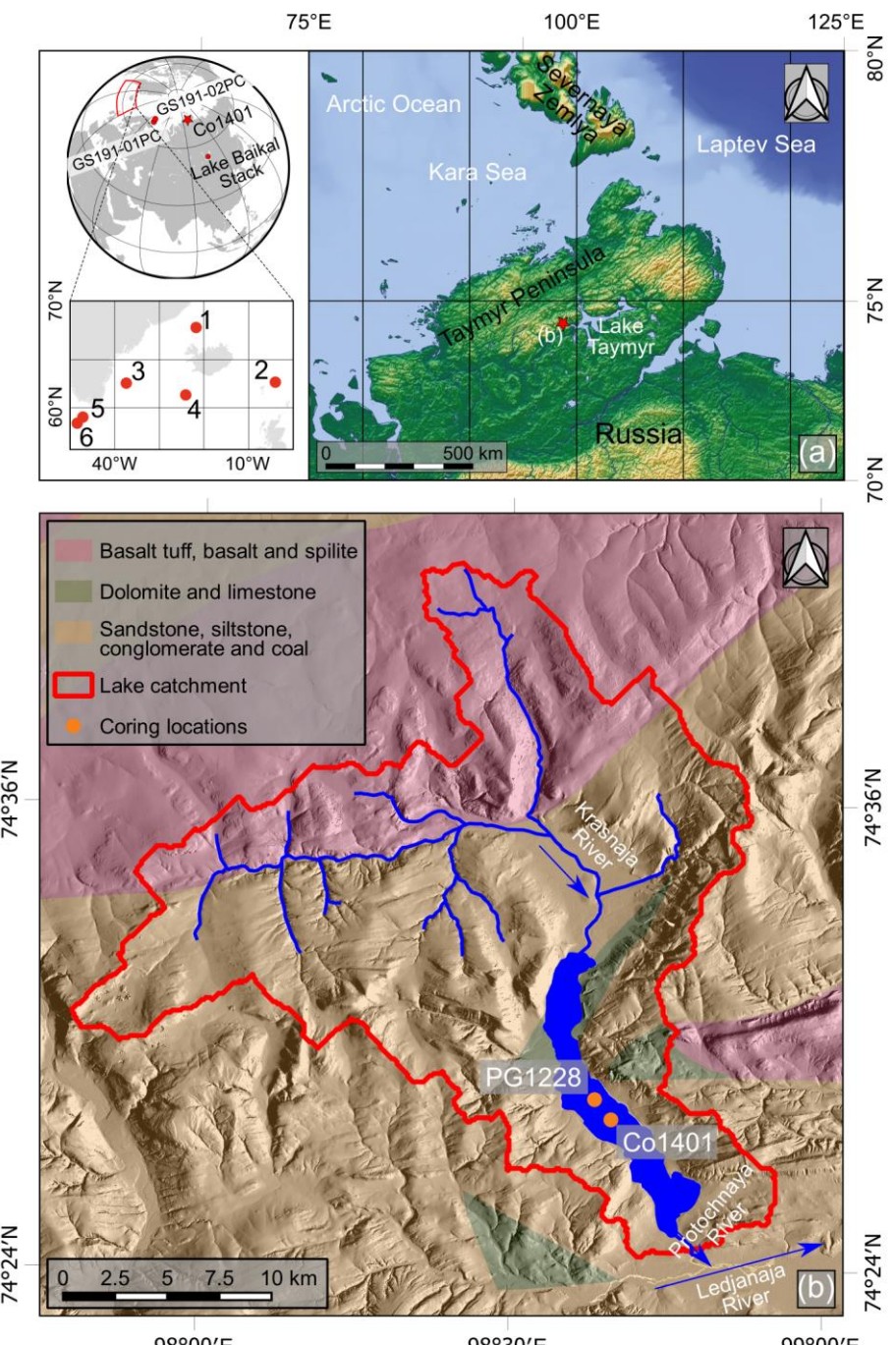

**Fig. 1: Locations of core Co1401 (red star) and of further records discussed in this study (red dots). (a) The map shows the location of Lake Levinson-Lessing (red star) to the west of Lake Taymyr on the northern Taymyr Peninsula in the Russian Arctic. At the left: records discussed in the text (red dots) named or abbreviated as follows: 1 - PS2644-5; 2 - MD95-2009; 3 - SU90-24; 4 - ODP site 984; 5 - P012; 6 - P013. Maps created using GMT 6 (Wessel et al., 2019). (b) Map of the coring sites Co1401 and PG1228 (orange dots) including the main geological units (Gromov et al., 2014) in the catchment area of Lake Levinson-Lessing. Topography is based on ArcticDEM (Porter et al., 2018).**

Lake Levinson-Lessing is located in the southern Byrranga Mountains on the Taymyr Peninsula in northern Central Siberia (74°25' to 74°32' N and 99°33' to 99°48' E; Fig. 1). It follows a tectonic trench, which was glacially reshaped in the Early Weichselian (Niessen et al., 1999). The lake basin is 15 km long, up to 2 km wide, and covers a surface area of approximately 25 km². With a maximum water depth of ~120 m, Lake Levinson-Lessing is the deepest lake of the Taymyr Peninsula. The

water column of Lake Levinson-Lessing is characterized by generally low nutrient and oxygen availabilities in winter times (Ebel et al., 1999).

The catchment of Lake Levinson-Lessing covers an area of ~500 km$^2$ (Fig. 1b). The main tributary is the Krasnaya River, which enters the lake at the northern shore. Additional water and sediment supply are provided by 20 ephemeral streams, draining the adjacent eastern and western slopes especially during snow-melt and intensive rainfall events (Bolshiyanov and Anisimov, 1995). The only outflow is the Protochnaya River in the south, which drains Lake Levinson-Lessing via the Ledjanaya River and Lake Taymyr, and the Lower Taymyr River into the Kara Sea of the Arctic Ocean.

The surrounding Byrranga Mountain range consists of weakly metamorphosed to unmetamorphosed Palaeozoic rocks that have been repeatedly folded and faulted (Anisimov and Pospelov, 1999; Torsvik and Andersen, 2002). The largest part of the catchment of Lake Levinson-Lessing is formed by Permian conglomerates, sandstones, and siltstones with intercalated coal beds (Gromov et al., 2014; Fig. 1b). Small outcrops of Upper Silurian and Lower to Middle Devonian carbonate-rich limestones and dolomites are restricted to the slopes along the central part of the lake`s catchment. The northernmost part of the catchment is built up of extrusive and intrusive rocks of the Taymyr igneous suite (Walderhaug et al., 2005), which reflect basaltic magmatism during the Upper Permian and Lower Triassic.

The modern climate on the northern Taymyr Peninsula is characterized by typical permafrost conditions with up to eight months-long severe winters (mean January temperatures -34°C) and short cold summers (mean July temperatures of ~6°C). The frost-free period is approximately 50 days per year (Siegert and Bolshiyanov, 1995).

## 3 Material and methods

### 3.1 Coring of Co1401 and core lithology

Co1401 is a composite core of 6.3 cm diameter that was retrieved from ~112 m water depth in the central part of Lake Levinson-Lessing (74°27'53.64"N, 98°39'58.03"E; Fig. 1b) in early 2017. The upper 134 cm of the core were recovered with a gravity corer, the remaining part consists of a succession of 24 non-overlapping cores. Each of the cores had a length of up to 2 m and were retrieved with a percussion piston corer (both UWITEC, AT) successively from the same hole. The cores were split into approximately 1-m-long sections directly after recovery. The composite record has a total length of 45.95 m, with gaps of 2 to 36 cm between individual piston cores. The gaps result from core catchers and exclusion of core sections due to core disturbances detected after opening the cores. Scheidt et al. (2021a) found that only the uppermost 38 m composite depth (mcd) are undisturbed, while the lowermost four piston cores seem to contain sediments that were successively collected before reaching the target depths, probably due to a technical malfunction. Therefore, only the upper 38 mcd of Co1401 are considered in this study.

The undisturbed part of the core consists of fine-layered, hemipelagic sediments, dominantly composed of silt. Frequent and irregular incisions of turbidites intercalate the background sedimentation and result in a widely continuous sediment accumulation punctuated by short-term peaks. Turbidite thicknesses are <8 cm with grain sizes up to fine sand at the base and clay at the top. However, most of the sediment sequence is finely layered without signs of bioturbation. Organogenic material, comprising terrestrial, and aquatic macro remains as well as charcoal in low concentration.

### 3.2 Whole-core, half-core, and biogeochemical measurements

Before the core segments were opened, whole-core palaeomagnetic measurements with alternating field (AF) demagnetization to peak fields of up to 15 mT were conducted at the Leibniz Institute for Applied Geophysics (LIAG) (Scheidt et al., 2021a). The cores were then split lengthwise at the University of Cologne, where the surface of one core half was cleaned to obtain

high-resolution line-scan images. Hereafter, XRF-scanning was performed at 2 mm resolution (exposure time of 20 sec) using an ITRAX core scanner (Cox Analytical Systems, SE) equipped with a 1.9 kV Cr X-ray source and a Si-drift detector in combination with a multi-channel analyser (voltage: 30 kV, amperage: 55 mA). Total carbon (TC) and total inorganic carbon (TIC) were measured on 345 samples of finely ground (<63 µm) sediment material using a DIMATOC 2000 (Dimatec Analysentechnik GmbH, DE). Total organic carbon (TOC) was calculated from the difference between TIC and TC.

### 3.3 Discrete sample palaeomagnetic analyses

The entire core composite was subsampled continuously in 2.1 cm intervals using 6.3 cm$^3$ plastic boxes. From the resulting sample set, 730 turbidite-free samples were selected for measurements at the cryogenic magnetometer laboratory of the Geological Survey of Norway in Trondheim (NGU). The magnetic susceptibility ($\chi$) was determined using a Bartington MS3 with an attached MS2B single-sample sensor (low frequency mode, 5 sec. measurement time). The natural remanent magnetization (NRM), anhysteretic remanent magnetization (ARM), and isothermal remanent magnetization (IRM) were measured using the horizontal SQUID SRM (2G Enterprises, US). NRM measurements were followed by stepwise AF demagnetization in peak fields of 5, 10, 15, 20, 25, 30, 35, 40, 50, 60, 70, and 80 mT using a 2G 600 AF Demagnetizer. Thereafter, an ARM (80 mT peak AF, 100 µT bias field) and finally a 900 mT IRM were imparted. ARM and IRM were both AF demagnetized using the peak fields specified above. NRM and ARM measurements were visually inspected and corrected for individual flux jumps or SQUID miscounts. The results of the IRM measurements were not corrected because single flux jumps or SQUID miscounts even after demagnetization is far below 1% and does not influence the palaeomagnetic interpretation.

The characteristic remanent magnetization (ChRM) was determined by principal component analyses (Kirschvink, 1980) of NRM demagnetization curves using the software Remasoft 3.0 (Chadima and Hrouda, 2006). Five to eight consecutive AF demagnetization steps between 15 mT and 80 mT were used to determine the ChRM. During the PCA the origin was usually weighted like a data point. However, 14 fits were anchored at the origin only when clustering of individual horizontal components prevented the determination of a meaningful direction (Scheidt et al. 2021b). Because the core is not oriented to the north, the first core section was rotated to adjust magnetic declination of the topmost sample with the present-day value at the site (~13°). Subsequently, the underlying cores were oriented one after the other to obtain a smooth declination record. The RPI was determined by the pseudo-Thellier method of Tauxe et al. (1995) using the partial NRM (pNRM) between 30 and 50 mT demagnetization steps, and magnetic susceptibility, as well as partial ARM (pARM), and partial IRM (pIRM) each between 30 and 50 mT demagnetization steps as normalizers.

The reliability of RPI proxies calculated from the ratios pNRM/$\chi$, pNRM/pARM, and pNRM/pARM has been evaluated using coherence analyses (Tauxe and Wu, 1990). For this purpose, raw profile data of pNRM and the RPI normalizers have been interpolated to regular intervals of 5 cm using a least-squares collocation (LSC) method (Moritz, 1978) with a $\sigma = 1$ m Gaussian correlation function. LSC fills data gaps without introducing spurious oscillations. Squared coherence has been calculated from spectrograms and cross-spectrograms of the resampled signals obtained with the multitaper spectral estimation method described in Park et al. (1987) and Vernon et al. (1991), using the first ten 6$\pi$ Slepian tapers. The squared coherence $\gamma^2(v)$ is a number comprised between 0 (no coherence) and 1 (full coherence) that measures the correlation between two signals as a function of the frequency $v$. The squared coherence is significant at a confidence level $\alpha$ if $\gamma^2 > 1 - (1 - \alpha)^{2/(f-2)}$, where $f = n - 2$ is the number of degrees of freedom for $n$ tapers (Chave and Filloux, 1985). In our case ($f = 8$), the 95% confidence threshold for $\gamma^2$ is 0.348.

### 3.4 Reference records

The locations of reference records are shown in Fig. 1.

#### 3.4.1 GLOPIS-75 and related records

GLOPIS-75 (Laj et al., 2004) was compiled with RPI data from 24 sites in the southern and northern hemispheres. including the sites PS2644-5, MD95-2009, SU90-24, ODP984, P012, and P013 (Stoner et al., 1995; Stoner et al., 1998; Channell, 1999; Laj et al., 2000). These high-resolution northern individual records from GLOPIS-75 are also used in this study. The age model of the GLOPIS-75 stack is based on the NAPIS record (Laj et al., 2000), which in turn relies on the GISP2 timescale (Grootes and Stuiver, 1999). GLOPIS-75 was recently updated using the GICC05 age model of the NGRIP ice core and K/Ar and $^{40}Ar/^{39}Ar$ ages of lava flows (Laj et al., 2014; Laj and Kissel, 2015). The names GLOPIS-75 and GLOPIS-75-GICC05 will be used to distinguish between the records with the original and the updated time scales, respectively.

#### 3.4.2 Marine records from the Barents Sea

The cores GS191-01PC and GS191-02PC originated from the southwestern margin of the Svalbard Archipelago in the Barents Sea (Caricchi et al., 2019). The 17.37 m and 19.67-m-long records cover the last ~42 ka and ~58 ka, respectively, and represent the most complete records available for a latitude (76°31' N and 77°35' N, respectively) similar to that of Lake Levinson-Lessing. The age models of the cores rely on AMS $^{14}C$ measurements, the identification of lithostratigraphic features (e.g., Heinrich events and meltwater events), and correlation with the GLOPIS-75 stack, as well as with the geomagnetic field models SHA.DIF14k (Pavón-Carrasco et al., 2014) and GGF100k (Panovska et al., 2018).

#### 3.4.3 The Lake Baikal stack

Peck et al. (1996) presented a stacked palaeomagnetic record over the last 84 ka from a sediment record of Lake Baikal (Siberia). Although Lake Baikal is located at 50°30' N, ~1500 km away from Levinson-Lessing Lake, it is the nearest lacustrine record extending beyond the LGM. The age model of the Baikal stack is based on AMS $^{14}C$ ages and the correlation of rock magnetic climate proxy data to the marine oxygen isotope record of Martinson et al. (1987).

### 3.5 Optically stimulated luminescence dating

Optically stimulated luminescence (OSL) dating was carried out on four samples at the Cologne Luminescence Laboratory of the University of Cologne. The sampled material was obtained from core catchers, which were pushed into opaque liners immediately after core retrieval to exclude light exposure. Samples for burial dose determination were processed under subdued red-light conditions and pre-treated following standard procedures to extract the 4-11 µm quartz fraction (e.g., Zander and Hilgers, 2013). Samples were pipetted on 9.8 mm steel discs and measured on a Risø TL/OSL reader with a $^{90}Sr/^{90}Y$ beta source delivering ~0.11 Gy s$^{-1}$ at the sample position. Luminescence signals were stimulated by means of blue LEDs and detected through a Hoya U340 filter. Equivalent dose measurements followed the SAR protocol of Murray and Wintle (2003) with an experimentally determined preheat at 220 °C for 10 sec and a cutheat at 200 °C. The appropriateness of the applied protocol was checked by means of preheat tests, dose recovery tests, and CW-curve fitting that indicate dose independence of preheat temperatures, reproducibility of laboratory doses and a dominant fast component. For each sample 10 aliquots were measured and passed standard SAR acceptance criteria. Since equivalent dose distributions with over-dispersion values of less than 5 % gave no indication of incomplete resetting of quartz luminescence signals, burial dose calculation was based on the central age model (Galbraith et al., 1999). The dosimetry was based on uranium, thorium, and potassium contents of the surrounding sediment derived from high-resolution gamma spectrometry measured at the VKTA Rossendorf. Measured water

contents were corrected for the effects of compaction during the burial period following Buylaert et al. (2013). Dose rate and age calculation were performed using the DRAC software version 1.2 (Durcan et al., 2015).

## 3.6 Radiocarbon dating

AMS [14]C dating on macro-organic remains was carried out on ten samples. Wherever applicable, macro remains were retrieved directly from the sediment core after core opening and during subsampling. In most cases, however, they had to be isolated from 1-4 g subsamples, which were sieved using mesh widths of 125, 63, and 36 µm. In the isolated fractions, larger organic remains were picked out by hand and smaller ones isolated by density separation. For the latter, a heavy-density solution consisting of sodium polytungstate (calibrated to 2.3 g cm$^{-3}$) was added to the samples. After 15 min of centrifugation at 2500 rpm, the lighter material floating at the top was pipetted off the solution. This step was repeated three times. Finally, the extracted material was rinsed using deionized water to remove and dissolve any residual sodium polytungstate. The isolated bulk organic and terrestrial plant remains were pre-treated according to the standard protocol for organic materials and converted to elemental carbon by graphitization (Rethemeyer et al., 2019). Samples were measured at the CologneAMS centre at the University of Cologne. Radiocarbon dates were calibrated to calendar years BP (AD 1950) using the IntCal20 calibration curve for terrestrial northern hemisphere material (Reimer et al., 2020).

## 4 Results and discussion

### 4.1 Basic chronology of core Co1401

The OSL (Table 1) and AMS [14]C (Table 2) dating results suggest a quite regular sediment accumulation with an age of ~60 ka at the base of the succession (Fig. 2). In a previous study of Levinson-Lessing Lake using sediment core PG1228, age discrepancies of ~2.5 kyr were reported between ages derived from Holocene pollen chronology and [14]C ages of macro remains, and attributed to redeposited organic material, inclusions of Permian charcoal fragments, and occasional sediment disturbances (Andreev et al., 2003). Although the [14]C ages in that study are consistent with the extrapolation of the pollen chronology trend from about 18 ka onwards, this study considers the possibility that the [14]C ages may be too old in core Co1401, particularly as OSL ages yield slightly younger ages than [14]C dating. Age differences between the two dating methods can be caused by hard-water effects on [14]C ages or other reasons mentioned in the work of Andreev et al. (2003). On the other hand, OSL results may be influenced by unaccounted changes of the water content, due to insufficient correction of sediment compaction, or incomplete luminescence signal resetting prior to sedimentation (Lang and Zolitschka, 2001; Buylaert et al., 2013). Age control by AMS-[14]C and OSL dating was therefore considered as a benchmark rather than a fixed age marker in this study.

**Table 1: Optically stimulated luminescence data and ages. Short names are used in the figures and discussion of this study. N – number of aliquots used for burial dose calculation, OD – over-dispersion, De – Equivalent dose, W$_{mod}$ – measured water contents corrected for the effect of sediment compaction, DR – dose rate. Ages are provided with 1σ uncertainties.**

| Short name | Lab-ID | Depth (mcd) | Mineral | N | OD (%) | De (Gy) | U (ppm) | Th (ppm) | K (%) | W$_{mod}$ (%) | DR (Gy/ka) | Age (ka) |
|---|---|---|---|---|---|---|---|---|---|---|---|---|
| O1 | C-L4563 | 9.94-10.00 | quartz | 10 | 0 | 47.7 ±0.6 | 3.0 ±0.4 | 10.3 ±0.7 | 2.40 ±0.15 | 61 | 2.9 ±0.2 | 17.6 ±1.0 |
| O2 | C-L4564 | 21.94-22.00 | quartz | 10 | 3 | 96.6 ±2.3 | 3.5 ±0.4 | 10.4 ±0.7 | 2.54 ±0.16 | 56 | 3.2 ±0.2 | 32.6 ±2.1 |
| O3 | C-L4565 | 27.94-28.00 | quartz | 10 | 5 | 126.9 ±3.9 | 3.3 ±0.5 | 9.4 ±0.6 | 2.30 ±0.15 | 50 | 3.2 ±0.2 | 45.1 ±3.1 |

| O4 | C-L4566 | 33.94-34.00 | quartz | 10 | 0 | 132.3 ±2.3 | 2.9 ±0.4 | 8.9 ±0.6 | 2.22 ±0.14 | 47 | 3.0 ±0.2 | 48.9 ±3.2 |

**Table 2: Radiocarbon isotope data and ages. Short names are used in the figures and discussion in this study. The macro-remains were too small for an unambiguous identification and may be of mixed aquatic and terrestrial origin. Calibrated ages are given with 2σ uncertainty ranges.**

| Short name | AMS ID | Depth (mcd) | Dated material | F$^{14}$C | $^{14}$C age (year BP) | Age (cal. year BP) |
|---|---|---|---|---|---|---|
| A | COL5806.1.1 | 0.85 | Macro-remains | 0.6759±0.0034 | 3150±40 | 3350±102 |
| B | COL5807.1.1 | 3.05 | Macro-remains | 0.5177±0.0041 | 5290±60 | 6094±169 |
| C | COL5808.1.1 | 6.45 | Macro-remains | 0.2111±0.0016 | 12500±60 | 14681±384 |
| D | COL5809.1.1 | 10.01 | Macro-remains | 0.0808±0.0013 | 20200±130 | 24276±525 |
| E | COL5810.1.1 | 14.80 | Macro-remains | 0.0751±0.0023 | 20800±240 | 25024±610 |
| F | COL5722.1.1 | 17.49 | Macro-remains | 0.0394±0.0018 | 26000±370 | 30159±767 |
| G | COL5723.1.1 | 22.30 | Macro-remains | 0.0200±0.0008 | 31400±310 | 35369±644 |
| H | COL5724.1.1 | 25.29 | Macro-remains | 0.0113±0.0017 | 36000±1200 | 40605±2195 |
| I | COL6876.1.1 | 33.08 | Macro-remains | 0.0022±0.0018 | 49900±880 | 52925±2056 |

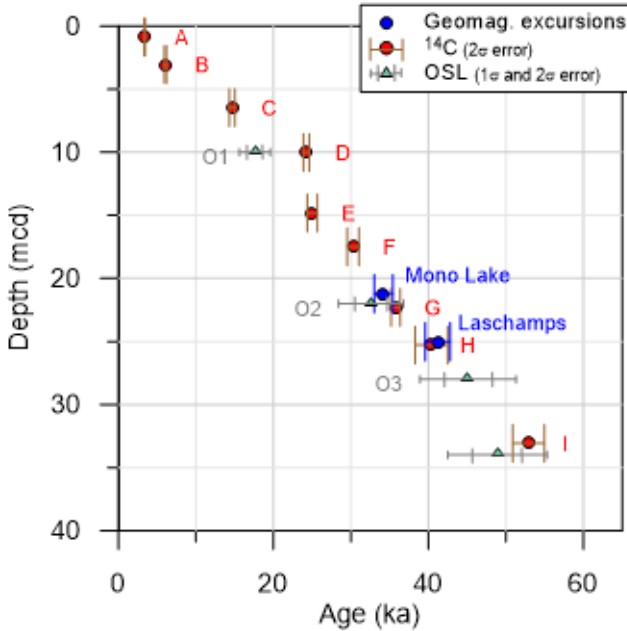

**Fig. 2: OSL and AMS $^{14}$C dating results of core composite Co1401, and locations of Laschamps and Mono Lake excursions. Ages and age uncertainties of the geomagnetic excursions are taken from Laj et al. (2014).**

**4.2 Magnetic mineralogy and selected geochemical properties**

The magnetic mineralogy of Co1401 was investigated in Scheidt et al. (2021a) to determine the suitability of this core for palaeomagnetic investigations. It has been shown that the sediment meets the basic requirements for good sedimentary RPI recorders as specified by Tauxe (1993) and Stoner and St-Onge (2007). Results suggest that the main remanence carriers of Co1401 are ≤5 µm pseudo-single domain (PSD) (titano-) magnetite and maghemite. Additionally, up to 8 % of the saturation remanent magnetization is carried by single domain particles. The contribution of multi-domain (MD) and superparamagnetic (SP) particles is generally low. The magnetic mineralogy appears to be exceptionally homogeneous between 6.7 and 38 mcd. Above 6.7 mcd, bulk hysteresis parameters suggest smaller magnetic grain sizes, but still in the PSD range. As documented

by first-order reversal curves, this apparent fining is due to the additional contribution of single domain greigite. The maximum contribution of greigite to the saturation remanence is negligible in the 0-20 mT range but may reach ~50% above ~50 mT at ~3 mcd. Based on these results, samples collected from Co1401 are generally well suited for RPI studies, although caution should be exercised in the upper meters.

In total, 90 samples were discarded (red dots in Fig. 3), for the following reasons:

First, sections with erratic, discontinuous ChRM directions were assumed to be affected by core disturbances. Second, samples at the end of core sections with significantly lower NRM values than adjacent samples were discarded because they were likely affected by oxidation of unstable remanence carriers (e.g., greigite) or intensity loss by deformation. Third, samples with particularly elevated NRM values (≥2 times larger than adjacent samples) or Fe/Ti ratios were assumed to be lithologically different from the surrounding samples. Such differences could be caused, for example, by turbidite horizons that were overlooked during sample selection. And fourth, samples with unstable demagnetization behaviour were considered unreliable.

Samples that passed the above selection criteria are characterized by consistently low and stable values in the Fe/Ti ratio (Fig. 3), which are generally interpreted as an expression of relatively stable redox conditions (cf., van der Bilt et al., 2015). It is important to note, however, that initial greigite formation in the uppermost meters is not detected by this proxy, because at the onset of iron reduction the smallest particles are affected first, which represent only a small portion of the total Fe content of the magnetic mineral fraction. Sample selection resulted also in a reduction of the already low scatter of NRM, $\chi$, ARM, and IRM values (Fig. 3). Interestingly, downcore variations of the overall low total organic carbon (TOC) content of Co1401 are anticorrelated with concentration-dependent magnetic parameters (Fig. 3). Since organic matter is more stable under conditions where ferromagnetic oxides become unstable, the observed anti-correlation could be related to the conservation potential of TOC and the magnetic mineral fraction and thus to the redox conditions prevailing in the lake (Snowball 1993). However, traces of sulphides could only be detected in the upper part of the core (Scheidt et al. 2021a). Therefore, increases and decreases in magnetic parameters (especially in the lower part of the core) are rather attributed to environmentally induced changes in the input of detrital magnetite than magnetite dissolution processes. In addition, in times of higher biological productivity vegetation density may have limited catchment erosion and thus, the supply of minerogenic sediments to the lake. Different sources of the deposited material are rather unlikely, since no variations of the lithogenic mineral magnetic composition has been detected (Scheidt et al., 2021a).

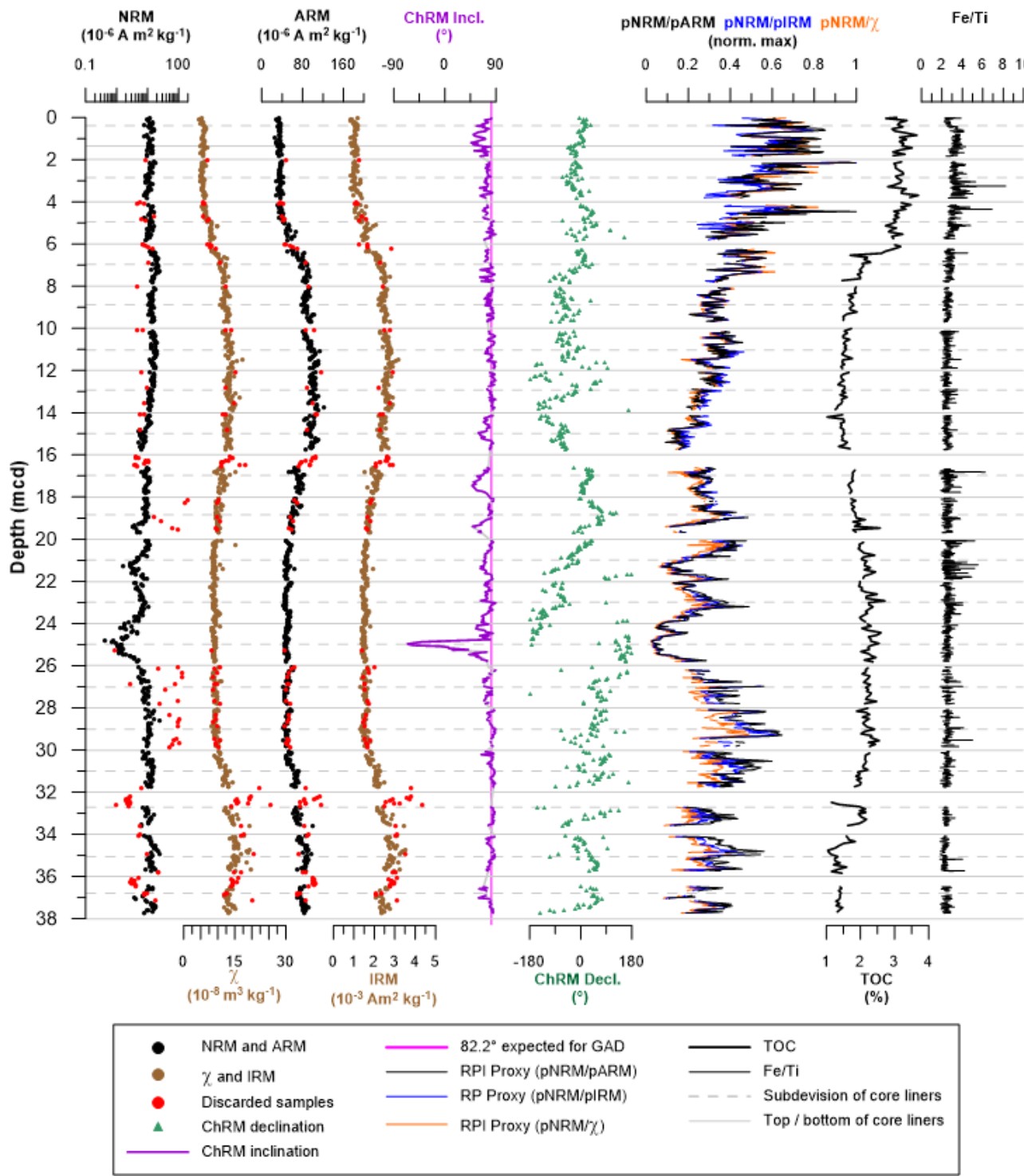

270

**Fig. 3: Downcore plots of magnetic and chemical parameters in core Co1401. Please note that the NRM is presented on a logarithmic scale. RPI proxies are normalized to respective maximum values. Discarded samples are not depicted in the ChRM inclination and declination, nor in the RPI proxies and Fe/Ti.**

### 4.3 Remanence acquisition in Lake Levinson-Lessing

275     The chronostratigraphic relevance of a palaeomagnetic sediment succession depends crucially on the NRM acquisition mechanism. Mineral magnetic analyses of core Co1401 do not suggest significant changes of the recording mechanism. Only in the uppermost part of the succession greigite formation has started. Consequently, it is possible that the magnetic signals of the individual samples from this part of the core integrate over the primary magnetization and a secondary magnetization that arose at any time. Since the greigite formation seems to be initial, we assume the main part of the magnetic signal to be primary.

However, this needs to be analysed in greater detail, so the surface sediments up to approximately 6.7 mcd should only be considered cautiously at present.

For the remaining part of the core, between 6.7 and 38 mcd, further comparison with other records require some considerations about the lock-in depth. The lock-in depth is defined as the depth below the sediment-water interface where a post-depositional remanent magnetization (pDRM) becomes locked and contributes to the permanent record. The occurrence and extent of this effect depends on several factors, including water content, grain size, sedimentation rate, and bioturbation intensity (e.g., Tauxe et al., 2006; Roberts et al., 2013; Egli and Zhao, 2015; Valet et al., 2017), but the underlying processes are poorly known. Therefore, the lock-in depth is described by an empirical lock-in function representing the fraction of total NRM that is blocked above a given depth (e.g., Nilsson et al. 2018, and references therein). The median and width of the lock-in function, divided by the sedimentation rate, give the mean delay and time constant of the recording process, respectively. A small delay is essential for establishing a chronology with small uncertainties.

Core Co1401 provides a rare continuity in TOC (Fig. 3), grain size, and water content in most parts of the core. Due to the fine-grained composition and soft consistency in the upper meters, a significant lock-in effect is expected. Since an accurate and precise chronology (e.g., varve chronology) for a systematic evaluation of the dimension of the lock-in effect is not available for Lake Levinson-Lessing, we refer to recent studies of varved sediments from two lakes in Sweden to obtain a rough estimate of the potential acquisition delay. Similar to our study, the age-depth relationships of lakes Kälksjön and Gyltigesjön indicate high and overall nearly constant sedimentation rates of approximately 59 cm ka$^{-1}$ during the past c. 3 ka (Mellström et al., 2015) and approximately 75 cm ka$^{-1}$ during the past 6 ka (Snowball et al., 2013), respectively. Furthermore, the preservation of sedimentary structures in these Swedish records largely excludes bioturbation, just as at Lake Levinson-Lessing. Yet, reported lock-in depths vary widely for both of the Swedish lakes. Mellström et al. (2015) state the lock-in depth to be between 30 cm and 80 cm for Kälksjön Lake, while Nilsson et al. (2018) modelled a half lock-in depth of 16.6±3.3 cm. A similar variance can be recognized for Gyltigesjön Lake, for which Snowball et al. (2013) specified a minimum estimate of 21-34 cm. By contrast, Mellström et al. (2015) modelled plausible lock-in depths between 50-160 cm for this lake, and Nilsson et al. (2018) yield a half lock-in depth of 30.7±6.5 cm by their approach. For comparison, it is important to note that the half lock-in depth can only be transferred into the lock-in depth by doubling of the value for symmetric lock-in functions (e.g., rectangular or Gaussian), but not for other shapes (linear, cubic or exponential). Also, the used geomagnetic field reference curves influence the final estimates, which is another reason for the different estimates of the lock-in depth of the Swedish lakes. Snowball et al. (2013) used the FENNOSTACK (Snowball et al., 2007) as reference curve, which is a stack of palaeomagnetic data of six varved lake sediment succession from Sweden and Finland and does not consider corrections for lock-in-depths in the underlying data sets. Using this reference, they obtained much lower lock-in minimum estimates than Mellström et al. (2015), who used four archaeomagnetic field models as reference, in which lock-in delays should not be an issue. Mellström et al. (2015) explain the appropriateness of their exceptionally deep lock-in depth estimates with the relatively low wet density and high organic content (45-50 % loss-on-ignition; LOI) of the lake`s sediment. Finally, the follow-up study of Nilsson et al. (2018) also used archaeomagnetic field models but additionally applied Bayesian modelling. They obtained similar lock-in depths as Snowball et al. (2013) and discuss the potentials and limitations of modelling, given the lack of high latitude data in archaeomagnetic field models. From this we can learn that, in addition to the lock-in function and the reference curves used, modelling approaches (programmes used, setting of parameters, and more) can also significantly influence the estimates of the lock-in depth. Due to the lack of high-resolution reference records, we will not model the lock-in depths in this work. However, if we agree with the assumption that TOC correlates positively with the lock-in depth and assuming a LOI to TOC ratio of approximately two (Vereş, 2002; Haflidason et al., 2019), the lock-in depths in Lake Levinson-Lessing must be well below those specified for both Swedish lakes (cf., section 4.5). Assuming a linear lock-in function, a lock-in depth range of 10-30 cm, and a sedimentation rate of 70 cm ka$^{-1}$ for Lake Levinson-Lessing, the expected time lag is of the order of 140-430 years and the signal can be expected to show corresponding smoothing effects.

## 4.4 Palaeosecular variations and relative palaeointensity

AF demagnetization to 80 mT removes, on average, 92.8 % of the NRM left after the initial 15 mT whole-core demagnetization. A small viscous overprint, which is fully removed after the 15 mT AF step, was observed only in a limited number of samples. Additionally, a few samples around 3 mcd acquired a gyroremanence in fields >60 mT, which is typically related to the presence of greigite (e.g., Scheidt et al., 2015). In the PCA, the ChRM was identified by use of 5 to 8 consecutive AF-demagnetization steps from the steepest part of the AF demagnetization curves. By using this part viscous overprints and increased influence of measurement noise at low magnetization are excluded and smallest maximum angular deviation (MAD) values are gained. In general, slightly higher AF fields were used in the upper part of the core than in the lower part (Fig. 4a, 4b). In case of steep inclinations, the horizontal component frequently clusters around or close to the origin (Fig. 4c). In almost all samples the 30 mT to 50 mT demagnetization steps were included in the determination of the ChRM, as this range was used to determine the RPI proxies. An exception was made for 16 samples from the depth range between 24.65 and 26.26 mcd, where 5 to 6 AF demagnetization steps between 15 mT and 40 mT were used. Here, the additional use of the 50 mT level would not have led to different directions, but only to much higher MAD values (see Scheidt et al. (2021b) for a detailed presentation of the steps used for individual samples). The MAD values of the complete dataset range from 0.2° to 7.6° with a mean value of 1.4° and a median value of 1.1°. Except for two samples from the upper 1.3 mcd of core Co1401, MAD values >5 were only determined for samples with low NRM values (40.9 $10^{-8}$ to 264.5 $10^{-8}$ Am$^2$ kg$^{-1}$) in the section between 24.19 and 25.53 mcd. It thus can be assumed that the higher influence of measurement noise at low magnetization values and a lower degree of alignment of the magnetic spins of magnetic particles in a weak geomagnetic dipole field caused samples around ~25 mcd to show larger directional scattering during AF demagnetization (Fig. 4d). Overall, one stable magnetic direction was extracted in all samples. The characteristic inclination varies between 89.7° and -66.1° (Fig. 3), with a mean and medium value of 75.9° and 77.5°, respectively (not considering the sections with lower and reversed inclination values around 25 mcd). The deviation to today's inclination (~86.9°; NOAA, (2021)) and the geomagnetic axial dipole (GAD expected at the site (82.1°) may be due to inclination shallowing, but could partly also result from sampling procedures. Due to the lack of orientation of the cores with respect to the north direction, and because the individual 2-m-long cores do not overlap, the consecutive rotation of each core section to make the declination curve continuous adds a cumulative error that increases with depth (error propagation). Thus, only relative changes of the characteristic declination within core sections can be considered reliable. In a few cases, sudden changes in declination also have been detected within cores. Here, the core parts were probably rotated against each other during transport along the layering around the Z-axis. As these movement planes were not found within samples, this rotation could not have affected the inclination or the RPI. However, the uncertainty in the declination is considered too large to calculate a VGP pathway for Co1401 or to compare its PSV with reference curves.

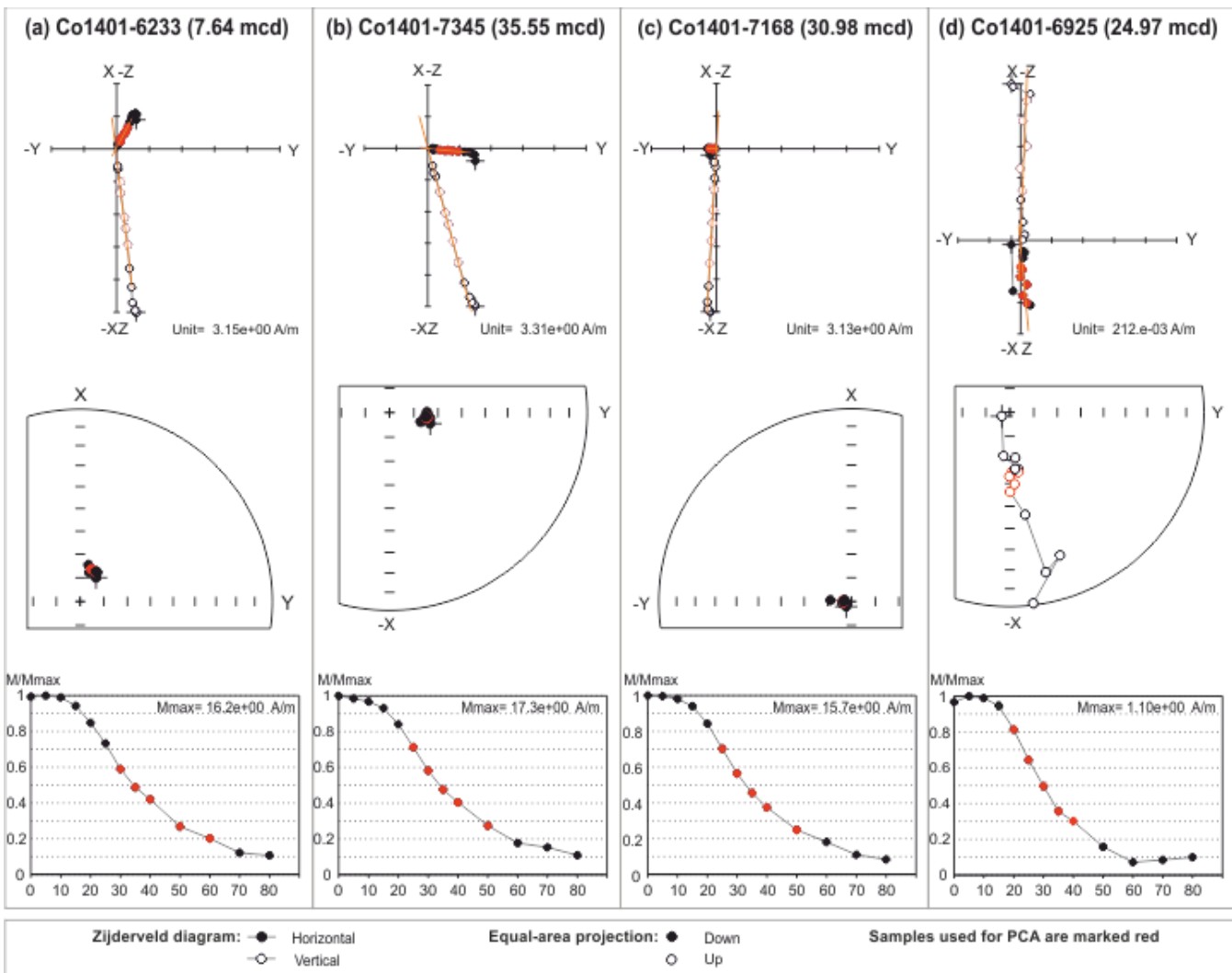

**Fig. 4: AF demagnetization characteristics of selected samples. Zijderveld diagrams, equal area projections and normalized NRM intensity are shown in the first, second, and third row, respectively. Points used to define the ChRM are marked in red.**

In addition to the directional information of the EMF, the detrital remanent magnetization of sediments is able to reliably store its relative strength (Banerjee and Mellema, 1974; Levi and Banerjee, 1976; King et al., 1983) given certain characteristics. Following Tauxe (1993) and Stoner and St-Onge (2007), the magnetization of favourable RPI recorders is mainly controlled by magnetite in PSD range, while the portions of SP and MD particles are small. In addition, the magnetic mineralogy should be homogeneous and variations of particle concentrations should be smaller than one magnitude. It was shown by Scheidt et al. (2021a) that this all applies to the sediments of core Co1401. As additionally requested for RPI determination, AF demagnetization was conducted and an uncomplicated behaviour of the directions during demagnetization with MAD<5 is shown in this study. AF demagnetization of ARM and IRM was carried out in the same steps as NRM, showing that the normalizers activate the same mineral fraction responsible for NRM. The RPI proxies $pNRM_{30-50mT}/\chi$, $pNRM_{30-50mT}/pARM_{30-50mT}$ and $pNRM_{30-50mT}/pIRM_{30-50mT}$ are strongly correlated with each other ($\gamma^2 > 0.75$), but not with their respective normalizers ($\gamma^2 < 0.37$), as shown by coherence analysis (Fig. 5). For comparison with other studies, the coefficient of determination $R^2$ is < 0.03 for all RPI proxies. This is expected if NRM normalization removes all remanence variations related to concentration and mineralogic changes of the remanence carriers. RPI proxies based on $\chi$- and IRM-normalization are practically identical over the whole core depth ($\gamma^2 > 0.93$), as expected from a constant $\chi$/IRM ratio. ARM yields a slightly different RPI proxy due to its selectivity towards SD remanence carriers. ARM appears to be the least effective normalizer if the whole core is considered, due to the larger coherence with its corresponding RPI proxy (Fig. 5f),

but all three normalizers perform similarly below 6.7 mcd ($\gamma^2 < 0.32$, Fig.5d-f). It thus appears that $pNRM_{30-50mT}/pARM_{30-50mT}$ is slightly affected by a minor greigite contribution above 6.7 mcd. This is confirmed by the relatively large coherence values observed between RPI proxies on the one hand, and the magnetic grain size proxies $\chi/IRM$ and $IRM/ARM$ on the other hand, but only if the greigite-affected depth interval is included (Fig. 5g-i). In this case, squared coherence values exceed the 95% significance threshold for certain frequency ranges, while $\gamma^2 < 0.32$ below 6.7 mcd. In summary, $pNRM_{30-50mT}/\chi$ and $pNRM_{30-50mT}/pIRM_{30-50mT}$ can be considered as reliable RPI proxies, especially below 6.7 mcd, owing to the very homogeneous magnetic mineralogy of this part of the core and the absence of greigite-related chemical NRM components.

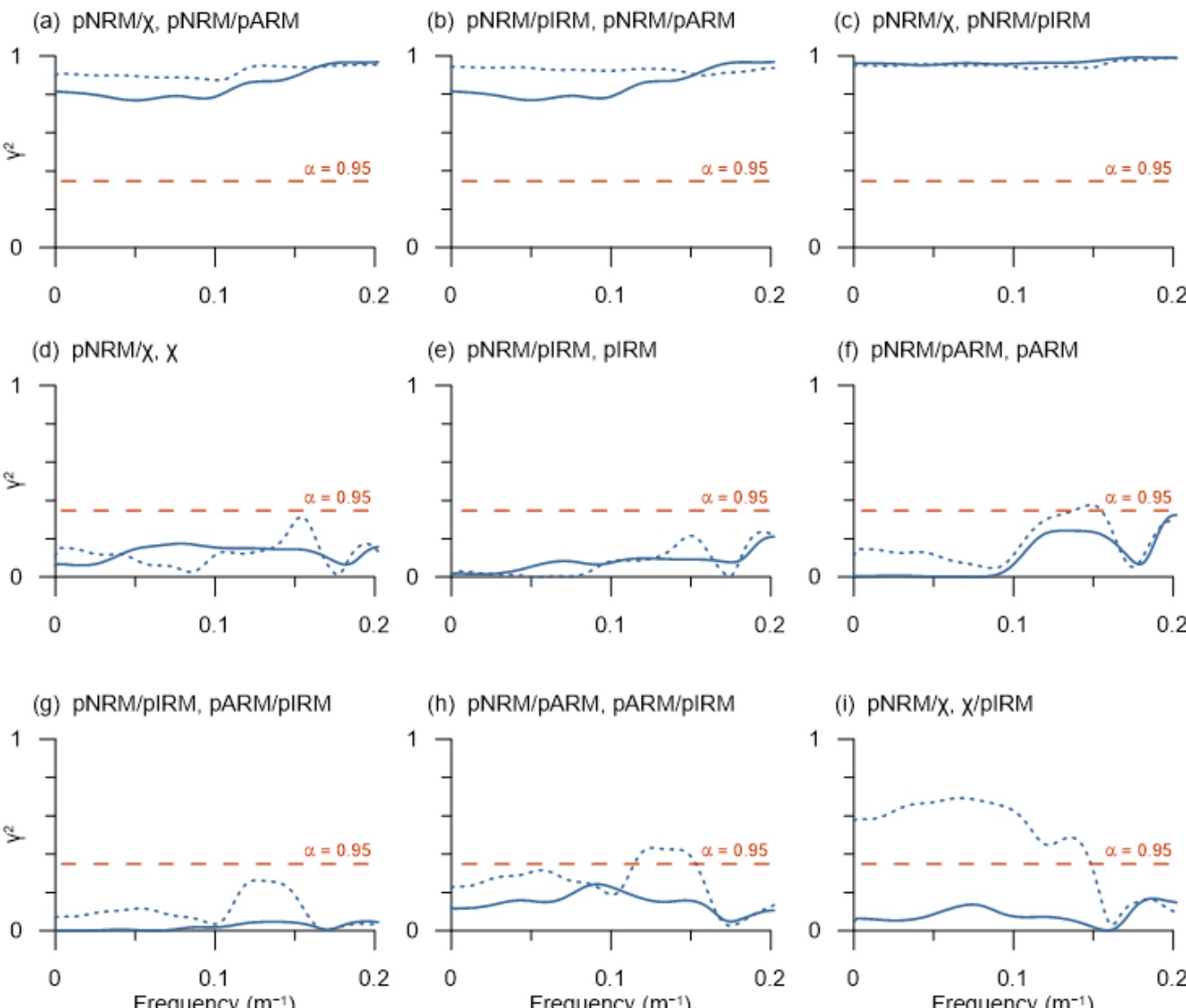

**Fig. 5: Squared coherence $\gamma^2$ between RPI proxies and rock magnetic parameters for the whole depth range (dashed blue lines), for >6.7 mcd (solid blue lines), compared to the 95% confidence level threshold (dashed red lines). (a-c) Squared coherence between RPI proxies. (d-f) Squared coherence between RPI proxies and the corresponding NRM normalizer. (g-i) Squared coherence between RPI proxies and magnetic grain size indicators.**

All RPI proxies show a clear decreasing trend between 0 and ~16 mcd, followed by two intervals with relatively constant average values at ~16-25 and ~25-38 mcd, respectively (Fig. 3). Short-term fluctuations superimpose these trends. The lowest RPI proxy values occur around 25 mcd, where reversed inclination is observed. The highest values are in the upper part of the core. Although all normalizers show a consistent trend, the intensities of the different normalizers change slightly relative to each other. Between 38 and ~19 mcd, $RPI_{ARM} > RPI_{IRM} > RPI_{\chi}$, with the differences between ~19 and ~26 mcd being smaller than below. However, between ~11 and ~9 mcd is $RPI_{IRM} > RPI_{ARM} > RPI_{\chi}$ and in 7.4-0 mcd is $RPI_{\chi} > RPI_{ARM} > RPI_{IRM}$, while

in between all normalizers lead to almost the same result. These small changes in the intensity of the normalizers relative to each other are probably due to naturally occurring, tiny variations in the composition and/or magnetic grain size of the ferromagnetic mineral fraction in these sections. Since the different RPI proxies show the same trend even in the upper part, where small amounts of greigite were detected, we assume that the RPI proxies reliably represent the variations of the EMF. The increases in fluctuations and total values of the normalized NRM from about 7.4 mcd upwards, and especially in the upper ~6 mcd, are probably due to a change in recording efficiency related to a change in magnetic mineralogy reported at about 6.7 mcd (Scheidt et al., 2021a). Alternatively, or in addition, a misorientation of magnetic minerals caused by compaction of the sediment in the lower part could result in lower RPI-proxy values.

Close to $^{14}$C age H at 25.29 mcd (40.6±2.2 ka), the lowest RPI-proxy values of the complete core are displayed and coincide with the negative values of the characteristic inclination (-66.1°). The age of $^{14}$C age H and the course of the RPI proxy curves coincide with those of the Laschamps geomagnetic excursion (Figs. 3, 6) reported by Laj et al. (2014) and Simon et al. (2020). Demonstrating replication with other regional records contribute further evidence that the normalized NRM reflects the RPI. Considering the Laschamps excursion at ~25 mcd, the Mono Lake excursion might be recorded above by either the RPI low at ~21.3 mcd or the one at ~19.4 mcd (Fig. 6). Both are associated with shallower inclination values (Fig. 3). Assuming a fairly regular sediment accumulation (cf., section 4.2; Ebel et al., 1999; Andreev et al., 2003), the RPI low at ~21.3 mcd represent the Mono Lake geomagnetic excursion. It is noteworthy that the course of RPI of Co1401 is very similar to that of GLOPIS-75-GICC05 VADM during the Mono Lake excursion (Fig. 6). The correlation scheme presented is also consistent with the findings of Korte et al. (2019), who analysed the Mono Lake excursion using global palaeomagnetic field models, and postulate that it consists of two epochs with moderate dipole lows located at about 34 and 31 ka. Simon et al. (2020) refers to models of Panovska et al. (2018) and Korte et al. (2019) and assume that the absence of Mono Lake in the beryllium ratio dipole moment (BeDM) stack (Fig. 6) may be related to the driving mechanisms of the Mono Lake, which are suggested to be governed by a non-dipole field pulse rather than by a significant dipole field collapse. Korte et al. (2019) also reports that transitional VGP latitudes (<45∘N) during the Mono Lake were only obtained from a few areas of the world, which are mainly located in the southern Pacific. It is thus not a contradiction to our assumption that the Mono Lake is recorded at ~21.3 mcd of Co1401 that the RPI low is only associated with shallower inclination values rather than true excursional directions. A lack of directional evidence for the Mono Lake excursion has also been reported in other studies carried out in the High North (e.g., Lund et al., 2017). The RPI low at ~19.4 mcd in Co1401 has been recognized similarly in the sediments of the Black Sea at about 31 ka (Jiabo et al., 2019) and, also softened, at about 28 ka in the core PS 2138-1 SL from the Arctic Ocean (Nowaczyk and Knies, 2000). The assignments of the RPI lows to the Laschamps and Mono Lake geomagnetic excursions shown in Figure 6 are within 2σ uncertainty of the absolute dating results. Finally, the lower inclination values at ~17.5 mcd (Fig. 3) do not correspond to lower RPI values. However, there are no correspondences with a large turbidite, a core end, or a change in the magnetic properties of the rock, nor is there any evidence that this element is a coring- or sampling-induced artefact.

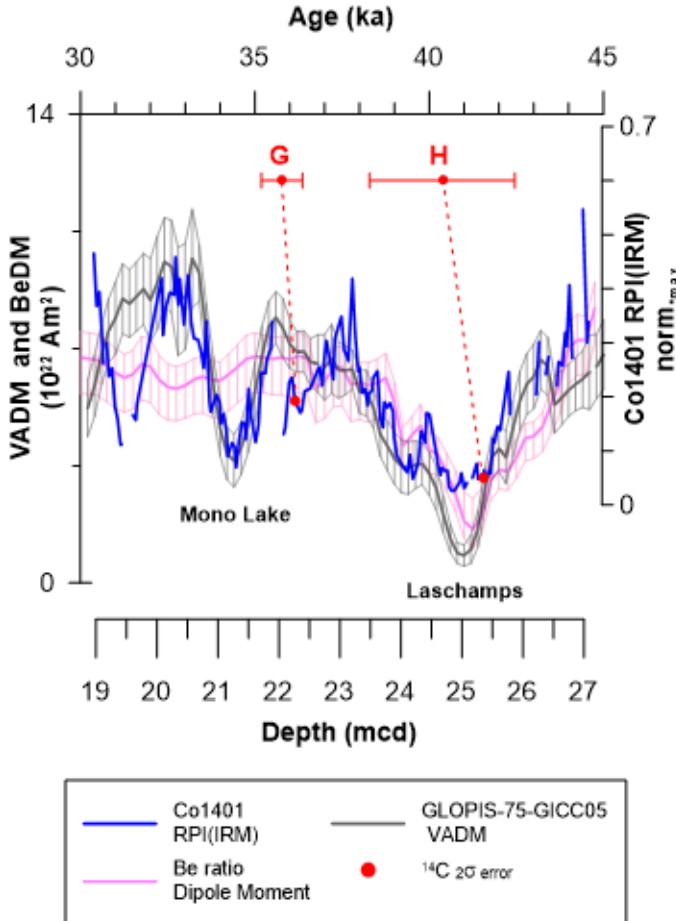

**Fig. 6: RPI(IRM) vs. depth of Co1401 (19-27 mcd) compared to GLOPIS-75-GICC05 VADM vs. age and the Be ratio dipole moment (BeDM) of Simon et al. (2020) vs. age (30-45 ka). The $^{14}$C ages H and G (Table 2) and the geomagnetic excursions Laschamps and Mono Lake are indicated.**

**4.5 Correlation to reference records and implications for Co1401**

Reference data for PSV correlation must come from nearby sites, are ideally characterized by continuous sedimentation at high rates, and should be supported by a robust age model. Unfortunately, there are no terrestrial or marine data with these properties within 1500 km of Lake Levinson-Lessing, which go back far enough in time. Overall, comparative records without severe glacial overprints are extremely rare across the entire Arctic Circle. Furthermore, not all records that fulfil the prerequisites for

palaeomagnetic analyses (Tauxe, 1993; Stoner and St-Onge, 2007) show (or have been measured with) sufficient resolution. Thereby, and because of core gaps leading to high uncertainty of the declination of Co1401, a robust evidence-based discussion of the variations of the EMF in the tangent cylinder using the PSV must wait until more data sets from the region are available. In this study we thus only compare and correlate the RPI data of core Co1401 to selected reference records (cf., section 3.4). In contrast to the PSV, the RPI is supposed to be a globally synchronous signal, though there is still uncertainty about the scale

over which the RPI records can be globally correlated (Channell et al., 2008). We have therefore selected reference data sets for correlation that (a) are as close as possible to Co1401, while (b) having a high resolution, and (c) covering the expected time of about 60-65 ka. As already stated above, such records are extremely rare in the Arctic Cycle. For this reason, we chose GLOPIS-75-GICC05 as master record, as it has the most recent age model. To refine the basic chronology originating from $^{14}$C and OSL dating (Fig. 2), we assume GLOPIS-75-GICC05 accurately represents the global variations in geomagnetic

intensity. It follows that by correlating the RPI(IRM) of Co1401 with GLOPIS-75-GICC05, its age model can be transferred to our record. In addition, six northern, high-resolution records included in GLOPIS-75 are used to demonstrate the variability

of the EMF in sedimentary recordings and to support our correlation scheme to GLOPIS-75-GICC05. As a result, we focus on the time interval covered by GLOPIS-75-GICC05, which is >10 ka (Fig. 7). We decided not to include additional reference records for the younger part of Co1401 for two reasons. Most importantly, there are a large number of Holocene RPI records that may serve as references and allow for many and complex approaches to the discussion. This would go beyond the scope of this publication and will therefore be published separately. On the other hand, we would like to complement the Holocene dataset with additional samples and carefully assess the effect caused by the presence of greigite. Notwithstanding, we tentatively establish a preliminary result for the range <10 ka by correlating the record to the virtual axial dipole moment (VADM) of GLOPIS-75-GICC05 (Fig. 7). Thereby we like to make the complete record available. This approach is reasonable due to the finding that magnetite is also the dominant phase in the upper meters of core Co1401 (Scheidt et al. 2021a) and the coherence is $\gamma^2 < 0.24$ for the RPI proxy based on pIRM. Nevertheless, the results of the upper part of the core should be considered preliminary. For the older part of the record, 12 additional tie lines were defined. Due to the natural variability of the EMF, the influence of sedimentological properties on records, and possible inaccuracies of the applied age models of the reference records, not all of these tie lines connect identically shaped features. Note that especially in the upper part of the core the use of $^{14}$C ages as fixed points did not lead to a coherent result. This is likely due to the possible age discrepancies of $^{14}$C ages (cf., section 4.1). For this reason, and because of the large uncertainties of the OSL ages, the $^{14}$C and OSL dating results were used as benchmarks when the RPI correlation allowed for different correlation schemes.

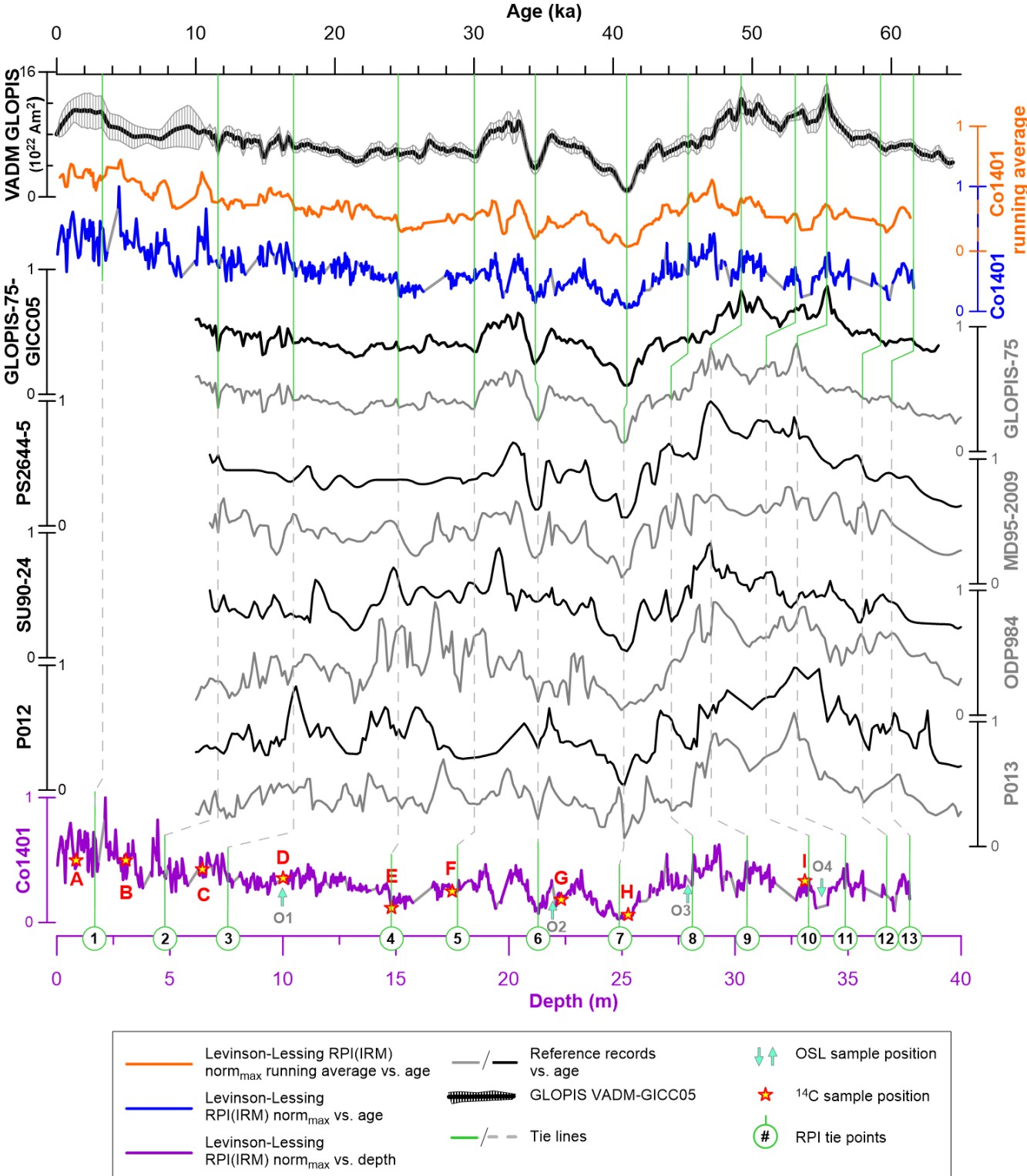

**Fig. 7: Tuning of the RPI(IRM) of Co1401 to six northern individual records of GLOPIS-75, and the GLOPIS-75 stack. Names of RPI records are given as axis labels. Please see section 3.4 for references. Please note age divergence between GLOPIS-75 and GLOPIS-75-GICC05. Running average of RPI(IRM) of Co1401 vs. age calculated with window width of 5 is shown for better identification of the general trend. All RPI data sets are normalized to the individual maximum values.**

The correlation procedure started with tie point (TP) 7 and TP 6 linking the RPI lows of core Co1401 and GLOPIS-75 that are associated with the Laschamps and Mono Lake geomagnetic excursions, respectively (Table 3, cf., section 4.4). In contrast to Co1401 and almost all of the individual sites shown in Figure 6, the RPI record of the GLOPIS-75 stack does not have pronounced drops in RPI between these two geomagnetic excursions. The [14]C sample G (35.78±0.59 cal. ka BP) provides an age for this decline in the RPI at Lake Levinson-Lessing. The OSL sample O2 (32.6±2.1 ka) is slightly younger than expected

from the age model but overlaps within its 2σ uncertainty (Fig. 2). The position of TP 5 in Co1401 was chosen to represent the RPI minimum below the pronounced RPI increase visible between ~30 and ~32 ka in GLOPIS-75 and MD95-2009, and to a lesser extend in P013 and PS2644-5. TP 5 was placed on the RPI low best supported by the [14]C sample F (30.26±0.77 cal. ka BP). Similarly, but starting from the RPI feature at ~15 m in Co1401, TP 4 was positioned in GLOPIS-75 considering [14]C sample E (25.01±0.66 cal. ka BP). If Co1401 would be solely correlated to GLOPIS-75, the minima below 15 mcd in Co1401 would likely be matched to the minima at ~21 ka in GLOPIS to delineate areas with similar trends (Fig. 7). In such a correlation scheme, TP 4 would be shifted to younger ages. The resulting age model (dashed line in Fig. 8a) would be supported by the age of OSL sample O1 (17.6±1.0 ka), but at least [14]C sample E and, depending on downward correlation, possibly [14]C sample F would be quite old. Therefore, we tentatively follow the correlation scheme that fits to the ages of the [14]C samples E and F, which is still within a 2σ error of the age of O1. This decision is supported by the similarity of the RPI of Co1401, MD95-2009 and SU90-24 in the section between TP 4 and TP 5.

The depth interval between TP 3 and TP 4 is the largest between the tie points, because no distinctive correlation features are present in Co1401. The age of [14]C sample D (24.26±0.37 cal. ka BP) was not considered, because admixture of tiny coal fragments has been detected, which likely caused erroneously high ages. The upward increasing trend in RPI visible between TP3 and TP4 in Co1401 is also present in GLOPIS-75, PS2644-5, MD95-2009 and SU90-24, albeit in a more attenuated form. TP 3 ties the depth level of Co1401 to the age of GLOPIS-75, at which the amplitude of variations gets larger towards the top. Beside GLOPIS-75, the course of the RPI curve of Co1401 between TP 3 and TP 2 is similar to P012, P013, ODP984, and SU90-24. TP 2 is set to an RPI minimum below the significant increase at~4.8 mcd. By this arrangement, the [14]C sample B (6.09±0.17 cal. ka BP) is slightly younger than the expected age (Fig. 8a). However, TP 2 and TP 1 are preliminary, as a detailed analysis of the varying magnetic carriers is necessary and will follow in a succedent study. Thus, we do not further discuss TP 2 and TP 1, nor the deviation from the age of [14]C samples B and A (3.35±0.10 cal. ka BP) suggested by the age model (Fig. 8a).

**Table 3: Tie points (TP) resulting from correlation of RPI(IRM) of core Co1401 with GLOPIS-75-GICC05.**

| Short name | Depth (mcd) | Age (ka) | Remark |
|---|---|---|---|
| TP 1 | 1.67 | 3.2 | Preliminary TP |
| TP 2 | 4.78 | 11.6 | Preliminary TP |
| TP 3 | 7.57 | 17.0 | |
| TP 4 | 14.79 | 24.6 | Position related to age of [14]C sample E |
| TP 5 | 17.73 | 30.0 | Position related to age of [14]C sample F |
| TP 6 | 21.30 | 34.4 | Mono Lake Geomagnetic Excursion; Supported by [14]C sample G |
| TP 7 | 24.88 | 41.0 | Laschamps Geomagnetic Excursion; Supported by [14]C sample H |
| TP 8 | 28.13 | 45.4 | Supported by OSL sample O3 |
| TP 9 | 30.53 | 49.2 | |
| TP 10 | 33.26 | 53.1 | Position related to age of [14]C sample I |
| TP 11 | 34.88 | 55.4 | |
| TP 12 | 36.68 | 59.2 | |
| TP 13 | 37.73 | 61.6 | |

Due to data gaps obscuring the course of the RPI of Co1401, the correlation with the reference datasets below TP 7 (Laschamps) was difficult. We decided to rely more on the age of [14]C sample I (52.92±2.06 cal. ka BP) than on the age of OSL

sample O4 (48.9±6.4 ka) because of the larger error of O4 and because of the close agreement of [14]C samples G and H with the ages constrained by the geomagnetic excursions. Thus, TP 10 was correlated to an RPI high supported by [14]C sample I before TP 8 and TP 9 were placed in the Co1401 to an RPI minimum and maximum, respectively. The resulting age-depth model is supported by the mean value of OSL sample O3 (45.1±3.1 ka). The course of the RPI of Co1401 between TP 7 and TP 10 is similar to all records shown in Figure 7. TP 11 assigns an age of 55.35 ka to the RPI peak in 34.88 mcd (Table 3).

The associated RPI feature is also expressed in PS2644-5 and ODP984 and appears to be slightly shifted to the younger in MD95-2009, P012 and P013 (Fig. 7). This correlation is still within a 2σ error of the OSL sample O4. Finally, TP 12 and TP 13 reflect a downcore decrease in RPI as observed in a number of reference records in Figure 7 Although the RPI between TL 11 and TL 13 of Co1401 appears to mimic that of SU90-24 and ODP984, the correlation was complicated due to large data gaps and is thus, associated with a certain degree of uncertainty.

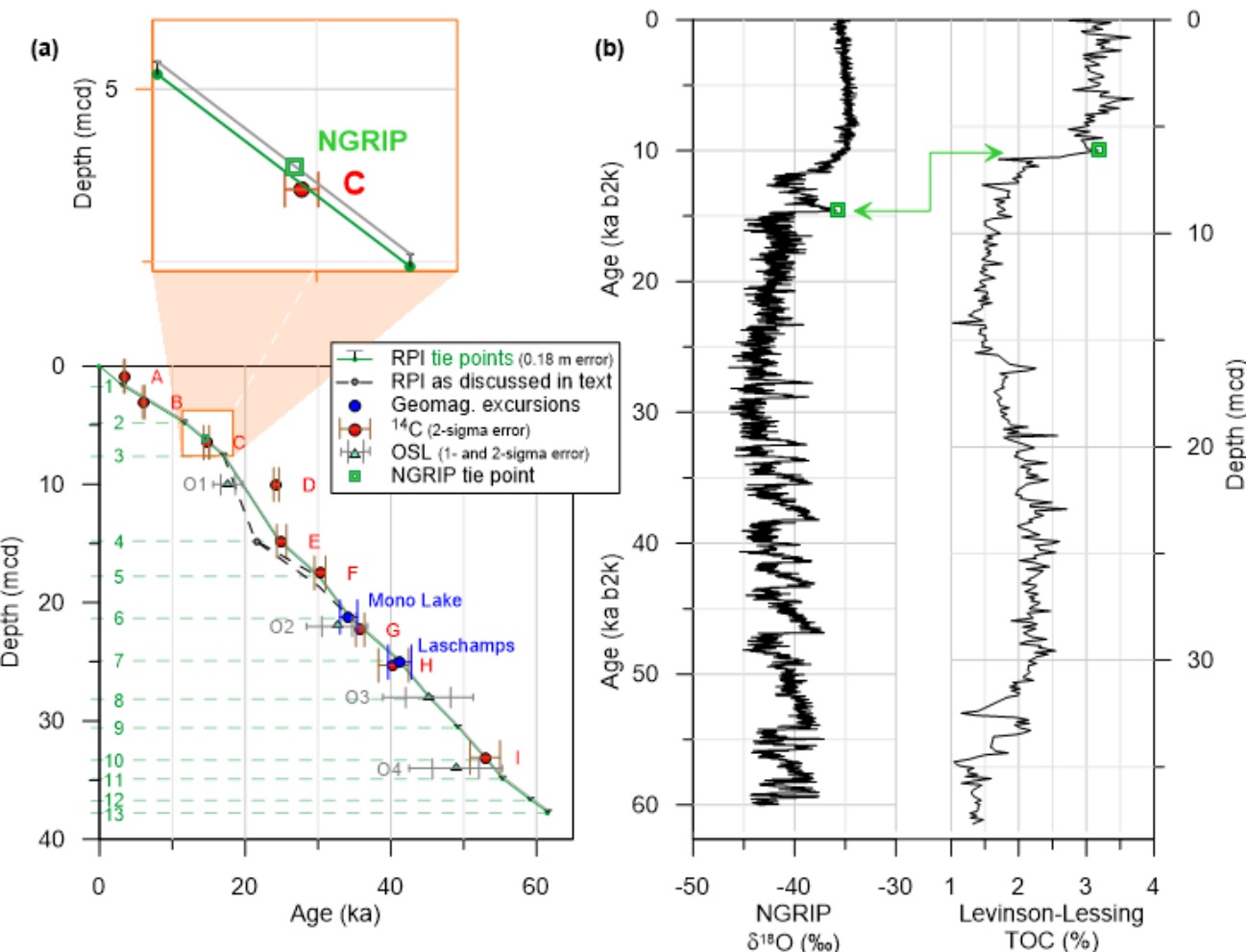

**Fig. 8: (a) Age-depth model of the core composite Co1401. Ages of the geomagnetic excursions shown after Laj et al. (2014). The grey line represents the age-depth relationship resulting of a constant lock-in depth of 18 cm. The age of the NGRIP tie point results from the comparison of δ[18]O vs. age and TOC of Co1401 vs. depth (b). Its age is recalculated to BP ages.**

The resulting age-depth model of Co1401 gives an age of ~62 ka at ~38 mcd (Fig. 8a). The age-depth model suggests two changes in sedimentation rates. From an average of 62 cm ka[-1] below 14.8 mcd, the sedimentation rate increases to 95 cm ka[-1] above, and becomes reduced again to 45 cm ka[-1] in the uppermost ~7.6 mcd (17 ka). The latter is similar to the outcome of previous studies on core PG1228 from Lake Levinson-Lessing, which suggested sedimentation rates between 46 and 75 cm ka[-1] for the Holocene (Hahne and Melles, 1999). For the complete core PG1228, a mean sedimentation rate of 70 cm ka[-1] (Ebel

et al., 1999) and a basal age of ~32 ka at ~22 m depth (Andreev et al., 2003) were determined. The somewhat lower average value of 62 cm ka$^{-1}$ in the sediment <38.8 ka in Co1401 may be explained by the more distal position of this coring site to the major sediment supply from the Krasnaya River (Lebas et al., 2019). However, it must be noted that the age model of PG1228 (Andreev et al., 2003) is based on uncalibrated ages for both, the dating of the macrofossil remains and humic acids, as well as of the definition of pollen zones and their correlation with the respective zones in other dated records. Thus, direct comparison of the ages from age model of PG1228 with the result of this study is not recommended. However, the offset of the $^{14}$C dates by several thousand years compared to the regional pollen chronology described by Andreev et al. (2003) seems to be independent of relative changes in the absolute age scale. A similar age deviation could not be found between the $^{14}$C ages and the palaeomagnetically-derived chronology. This is particularly evident from the consistency of the ages of the $^{14}$C samples G and H in relation to the geomagnetic excursions. Also, the lower sedimentation rates for an unknown duration sometime between about 17-8 ka BP, which is reported by Andreev et al. (2003) for PG1228, seems not to be expressed in Co1401. Instead, a slowdown of the sediment accumulation towards the top is supported by the correlation of TOC with the NGRIP δ$^{18}$O dataset (Svensson et al. 2008, and references therein). Based on palynological evidence provided by Lenz et al. (2022) and assuming that TOC responds without time delay to the change in global ice volume, an age of 14.6 ka (b2k) is assigned to 6.12 mcd (Fig. 8b). This Co1401-TOC to NGRIP correlation suggests a time lag resulting from a lock-in depth of ~18 cm (magnified section in Fig. 8a), though this is only a rough estimation. Uncertainties arise from the correlation schemes of Co1401-TOC to NGRIP and Co1401-RPI to GLOPIS-75, as well as from the influence of TP 2, which is only preliminary. It is also not known whether this value can be applied to sections at greater depths. Compared to the lock-in depth suggested for the Swedish lakes Kälksjön and Gyltigesjön (cf., section 4.3), the 18 cm estimated for Co1401 seems to be quite low. The smoothing effect associated with higher lock-in depths would explain the low variability of the RPI of Co1401 compared to the reference records with lower sedimentation rates. Thus, we suggest the value of 18 cm to be a minimum estimation for the lock-in depth at Lake Levinson-Lessing.

### 4.6 Implications for the High Arctic

During the Holocene, the geomagnetic field in the High Arctic exhibits important differences in its PSV and RPI behaviour compared to midlatitude and Low Arctic records (e.g., St-Onge and Stoner, 2011; Lund et al., 2016). In contrast, hardly any conclusions can be drawn about the behaviour of the EMF in the High Arctic during the Late Pleistocene. The few available records covering this period mainly originate from the central Arctic Ocean and are affected by sedimentation rates of <8 cm ka$^{-1}$ (e.g., Nowaczyk et al., 2003; Polyak et al., 2009). In addition to the resulting low temporal resolution, the interpretation of the palaeomagnetic evidence of these sites was complicated by inconsistencies in PSV and RPI. As explanations for these phenomena, magnetic excursions and diagenetically-induced self-reversals during maghemitization were discussed (e.g., O'Regan et al., 2008; Xuan and Channell, 2010). Just recently, however, the magnetic records of the Arctic Ocean have been shown to be strongly influenced by diagenetic processes that vary in intensity (Wiers et al., 2019; Wiers et al., 2020). This may call into question a number of palaeomagnetic data from marine sites of the Arctic and further thins out the available data. The continuous and high-resolution sediment record of core Co1401 thus, represents an exceptional archive for palaeomagnetic and palaeoenvironmental research. A recent study in the Barents Sea has provided another data set that does not report problems like those in the Arctic Ocean (Caricchi et al., 2019; Caricchi et al., 2020). However, the comparison of the RPI record of Co1401 to the respective sites (GS191-01PC and GS191-02PC) unveil large differences in the amplitude of RPI variations (Fig. 9). The overall significantly lower variability in the Levinson-Lessing Lake record may be due to the differing sedimentary processes. In contrast to the largely uniform lithology, overall constant accumulation and low variability in grain-size spectra of Co1401, the marine sediment cores from the Barents Sea are characterized by high variability with respect to all of these properties. The sedimentation rates of GS191-01PC and GS191-02PC are reported to average 46 and 29 cm ka$^{-1}$, respectively, but the lithology includes bioturbated mud, massive silt layers, and layers with sometimes massive concentrations

of pebbles and cobbles (ice rafted debris; Caricchi et al., 2019). Nevertheless, most parts below the Mono Lake geomagnetic excursion in Co1401 (marked in royal blue and black in Fig. 9) correlate well with those in GS191-02PC. As in most individual northern Atlantic records (Fig. 7), the Barents Sea records indicate an RPI decrease between the Laschamps geomagnetic excursion and a data gap obscuring the Mono Lake excursion. Accordingly, this seems to be a characteristic of the EMF at high northern latitudes, and the feature was averaged out by the global distribution of records combined in GLOPIS-75. Towards younger ages, a correlation of the Barents Sea records and Co1401 is prevented by the uneventful course of the RPI of Co1401. Only the general increasing trend can be recognized to some extent in the parts of the records coloured in purple (Fig. 9). The sections highlighted in light blue of GS191-01PC and GS191-02PC are characterized by strong loss of RPI intensity. This is opposed to the general trend represented by VADM of GLOPIS-GICC05 and could be related to diagenetic overprinting or a change in the composition of the mineral magnetic influx.

Compared to the Barents Sea records, the Baikal Lake stack shows a much lower RPI variability (Fig. 9). Despite the differences in the assigned ages of the Baikal Sea stack, clear analogies to Co1401 are recognizable. This is remarkable, since Peck et al. (1996) report average sedimentation rates as low as 13 cm ka$^{-1}$. Despite the 4.7-fold higher average sedimentation rate in Core 1401 of Levinson-Lessing Lake, the characteristics of the RPI do not appear to be better defined or more pronounced. This observation holds for the comparison with all colour-coded sections of the Baikal stack in Figure 9, though to a lesser extend in the part marked in black, in which the sedimentation rates of the Baikal stack are particularly low.

The overall strong similarities between Co1401, the GLOPIS-GICC05 VADM, and the Baikal stack of low temporal resolution from 50° latitude suggest that the dipole component of the EMF dominates the RPI variation at the Levinson-Lessing site. It remains an open question whether the latitude of Levinson-Lessing Lake of about 74°3' is not close enough to the magnetic North pole to record significant non-dipolar features expected inside the tangent cylinder. As mentioned earlier, gradual lock-in of magnetic minerals over long periods of time would also be a plausible explanation for the low variability of the RPI. However, considering the determined sedimentation rates, a uniform lock-in depth of 18 cm and a corresponding lock-in width of the half of the lock-in depth, we gain a smoothing constant <220 years, which would probably not erase all RPI variations. Further insights into the magnetic field on the tangential cylinder may arise from future studies combining the results of this study with RPI and PSV data from other sites (e.g., Minyuk and Subbotnikova (2021), as soon as palaeomagnetic data is made available), or from modelling the behaviour of the EMF in the Eurasian Arctic using the data presented.

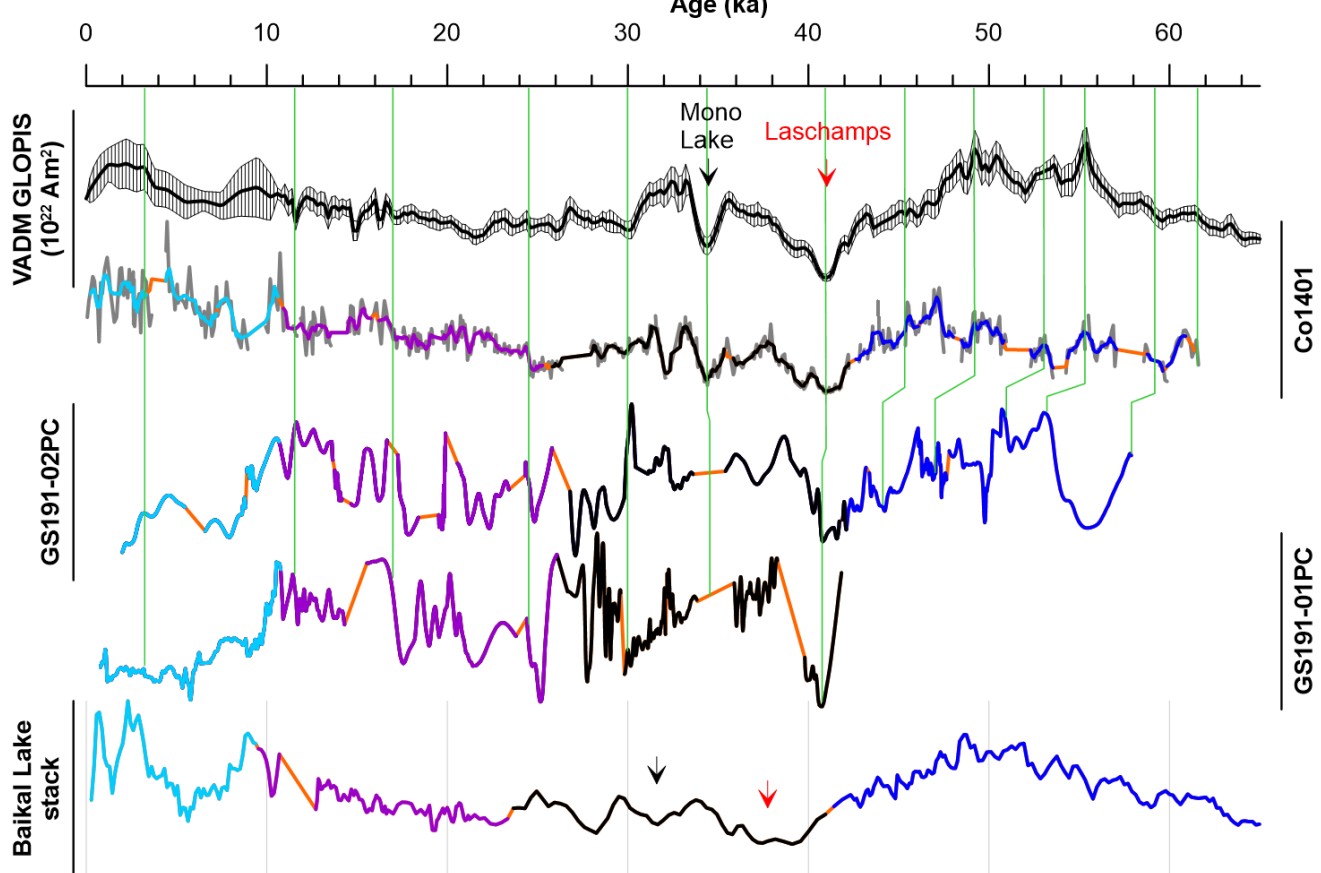

**Fig. 9:** Comparison of the smoothed RPI(IRM) of Co1401 (non-smoothed data in grey colours in the background) with RPI records from Lake Baikal (Baikal Lake stack) and the Barents Sea (GS191-01PC and GS191-02PC) and the VADM of GLOPIS-75-GICC05. All records are normalized to the respective maxima. For discussion, the RPI records are divided into four parts by colour code (light blue, purple, black, and royal blue). Gaps in the individual records are marked in orange. Please note the different ages assigned to the RPI characteristics of the different datasets, leading to a shift of the patterns against each other (c.f., section 3.4). Tie lines are shown as in Fig. 6. For the Baikal Lake stack, only the main grid lines are provided, as correlation is not readily possible. The locations of geomagnetic excursions are indicated by tie lines and arrows.

## 5 Conclusions

We conducted a high-resolution study of the RPI of composite core Co1401 from Lake Levinson-Lessing using ARM, IRM, and χ as normalizers and obtained consistent results throughout the upper 38 mcd of the core. The results for the upper 6.7 mcd indicated a change in magnetic mineralogy including initial greigite formation and the RPI fluctuations substantially increase in the upper part of the core. The provided correlation of the data <10 ka is thus preliminary. The lower part of the core (6.7-38 mcd) yields reliable RPI estimates owing to the very homogeneous magnetic mineralogy and the lack of greigite-related chemical NRM components. Palaeomagnetic events recorded by Co1401 include recordings of the geomagnetic excursions Laschamps and Mono Lake. The age-depth model for core Co1401 was established by a combined approach using the correlation of the RPI(IRM) to the GLOPIS-75 stack at the GICC05 time scale, AMS $^{14}$C, and OSL dating. The age-depth model suggests an age of ~62 ka at ~38 mcd. Average sedimentation rates shift from 45 cm ka$^{-1}$ between 0 and ~7.6 mcd (17 ka), via 95 cm ka$^{-1}$ between 7.6 and 14.8 mcd, to 62 cm ka$^{-1}$ below 14.8 mcd. The lock-in depth was tentatively set to 18 cm based on a single tie point obtained from correlation of the downcore changes in TOC with the NGRIP δ$^{18}$O record. This value suggests a smoothing constant of <220 years for the complete sedimentation record. Although the course of the characteristic declination is obscured by the lack of overlap of core segments of core Co1401, the PSV data may be used in the future, as soon as data of the general course of the declination from close-by study sites become available. At present, the

lacustrine sediment core Co1401 provides the only high-resolution RPI and PSV dataset extending to ~62 ka within a radius of more than 1500 km around Lake Levinson-Lessing.

## Acknowledgments

This study was funded by the German Federal Ministry for Education and Research (BMBF) within the scope of the project PLOT (Paleolimnological Transect, grant nos. 03G0859A and 03F0830A). Datasets for this research are available in Scheidt et al. (2021b). Special thanks are due to Richard Niederreiter, Sverre Aksnes, Andrej Andreev, Anna Cherezova, and Maxim V. Rukhkin for their competent help in retrieveing core Co1401. We are also very indebted to Carlo Laj and Cathrin Kissel for providing the current GLOPIS-75 data and the data of the individual GLOPIS-75 records.

## Data availability statement

Datasets for this research are available in Scheidt et al. (2021b).

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
