# Peer review of "A 62-ka geomagnetic palaeointensity record from the Taymyr Peninsula, Russian Arctic"

_Geochronology, 2021_

## Author Response (AR1)

Dear reviewers and editors,
In the interactive discussion we provided point-by-point answers to the reviewer's comments including descriptions of actions made, and reasons for our decisions, and answers to further questions. We have therefore decided to list here only the changes made, including references to the lines in the updated manuscript.

**Changes made related to Reviewer#1's suggestions**

**Action related to comment on line 28:**
Line 28/29: changed to also presents preliminary
**Action related to comment on line 50:**
Line 51: age and reference of the age was included.
**Action related to comment on line 95:**
Changed in lines 97, 279 and 296 as suggested.
**Action related to comment on line 99:**
Line 101: The word is deleted
**Action related to comment on line 101:**
The position of lines in Fig. 3 (=former Fig. 2) were corrected. We added information in section 3.1
**Action related to comment on line 140:**
Lines 146/147: The sentence is changed.
**Action related to comment on line 142:**
The complete section is reorganized following the suggestions of Reviewer#2. The related text passage is now part of section 4.4.
**Action related to comment on line 151:**
Lines 162/163: The references of the individual records are presented.
**Action related to comment on lines 151-167:**
Lines 160-179: The bullet points were changed to sub-sections.
**Action related to comment on line 211:**
We discarded the results of one sample with very different declination directions and to keep the complete data of the sample that shows only somewhat misaligned directions. Accordingly, the text sections were corrected.
**Action related to comment on line 263:**
Line 340f: We revised the sentence following the suggestions of reviewer#2.
**Action related to comment on line 276:**
We revised section 4.4 (=former section 4.3) and included new evidence. Until the proof is provided that data represents the RPI, we use other terms.
**Action related to comment on line 278:**
Line 389f: We make now aware of the change in normalizer intensity and discuss the implications of this issue.
**Action related to comment on line 288:**
Line 403ff: We explain the reasons for the assignment of the Mono Lake excursion to the RPI low at 21.3 mcd in greater detail.
**Action related to comment on line 297:**
We changed the order of the sections.
**Action related to comment on line 336:**
We explained now in greater detail in section 4.1 (former section 4.2) why the radiocarbon dates are not used as fixpoints.
**Action related to comment on line 357ff:**
Lines 469ff: We checked the text thoroughly and changed the wording. We think it is clear now that the RPI low is meant that is dated by 14C date G.
**Action related to comment on line 419:**
We have to disagree with this statement, since 8a (= former 6a) occurs for the first time in line 478 (in the previous version line 379), while Fig. 8b (= former 6b) is mentioned for the first time in line 532 (in the previous version line 432).
**Action related to comment on line 469:**
Line 580: We agree change the wording accordingly.

**Changes made related to Reviewer#2's suggestions**

**Action related to comment on the abstract:**
> Line 25ff: We corrected this issue in the short summary, the abstract, and in the introduction.

**Action related to comment on line 40:**
> Line 40: We use the wording as suggested by Reviewer#2. For the remaining issue of this comment, please see our answer in the interactive discussion.

**Action related to comment on line 46:**
> Please see our answer in the interactive discussion

**Action related to comment on line 134**
> Please see our answer in the interactive discussion

**Action related to comment on Section 3.4**
> To follow the advice of Reviewer#1 and #2 we will reorganize the order of the sections in the Results and discussion chapter as follows:
> > 4.1) Basic chronology (spanning a rough time frame)
> > 4.2) Magnetic mineralogy (shows the suitability of the sediment for RPI)
> > 4.3) Remanence acquisition (introducing the lock-in issue; position of the section was suggested by Reviewer#1)
> > 4.4) PSV+RPI (description and discussion of the data)
> > 4.5) Correlation (Discussion on reference records, correlation of the data and final age-depth model)
> > 4.6) Implications
> ❖ We shifted the part of section 3.4 which might be recognized as part of the discussion to section 4.4 (former section 4.3).
> ❖ Line 162: The missing references were included.
> ❖ Line 444: We included a statement in which we assume GLOPIS to precisely show the variations of the EMF. Thereby the age-depth model of GLOPIS and the related records shown can be transferred to Co1401
> ❖ Line 165/166: We corrected the information on K/Ar or Ar/Ar ages related to GLOPIS.

**Action related to comment on line 210:**
> We discarded the results of one sample with very different declination directions and to kept the complete data of the sample that shows only somewhat misaligned directions. Accordingly, the text sections were corrected.
> We deleted the part of the sentence to avoid ambiguities.

**Action related to comment on line 259f:**
> We revised the section to make sure all readers understand the mechanism we propose.

**Action related to comment on line 255:**
> Lines 323f: We recalculated the PCA of a number of samples using more steps (without obtaining other results). The new values are used now in the manuscript. We explain now in greater detail which steps are used and for which reason. The MAD is now included.

**Action related to comment on line 261:**
> We include a recent value of the inclination (line 343). Today's declination is mentioned in line 143. We state now only the mean and medium inclination values excluding the geomagnetic excursion.

**Action related to comment on line 266:**
> Line 344f: We included a short explanation and the keyword "error propagation" in brackets.

**Action related to comment on line 268:**
> Line 349: We clarified that the movement planes are not situated in the samples.

**Action related to comment on line 269:**
> Line 351: We corrected the term as suggested.

**Action related to comment on line 275:**
> We revised section 4.4 (=former section 4.3) and included new evidence. Until the proof is provided that data represents the RPI, we use other terms.

**Action related to comment on around line 277:**

Please see the Action related on the comment to line 375 below

**Action related to comment on 284f:**

Line 400ff We included an additional figure (Fig. 6) that shows the low normalized remanence intervals; i.e. Laschamps and Mono Lake and re-addressed the fact that 14C age H is close to these intervals. We completed the discussion regarding the Mono Lake.

**Action related to comment on line 335:**

Lines 429ff: Following Reviewer#2's suggestions, we rephrased the entire section4.5 (former 4.4).

**Action related to comment on line 355ff**

Line 403ff: We explain the reasons for the assignment of the Mono Lake excursion to the RPI low at 21.3 mcd in greater detail and corrected the wording as suggested.

**Action related to comment on around line 375:**

We made changes in all text sections related to the Holocene and greigite to make the reader understand that we do not expect this part of the core to be unreliable nor unsuitable to the method. We highlight now the much larger opportunities for discussion of the Holocene part in an own extensive study, which is actually already on its way.

---

## Author Response (AR2)

Dear editor,
We thank reviewer#1 for reviewing the manuscript again. Please find our replies to reviewer#1's suggestions and questions below:

Line 31: The authors say in the abstract that there are high lock in depths. Later in the manuscript this is discussed as being at least 18 cm, but, due to high sedimentation rates, this only represents a few hundred years. What do the authors mean by high lock in depths? 18cm is within the 'usual' range of 10-30 cm that is often quoted. A few hundred year delay is not a lot (marine sediment cores can have much larger offsets due to low sedimentation rates). Maybe the authors should think about their wording in this context, especially as it is in the abstract.

 Reply: Thank you for highlighting this issue. The wording is incorrect indeed. We changed the sentence into: "The low variability of the magnetic record compared to data sets of reference records with lower sedimentation rates may be due to a smoothing effect associated with the lock-in depths" to mention both, the low variability and the lock-in in the abstract.

Line 142: For transparency, I would mention how many samples of those used were anchored to the origin. This can have a large (positive) effect on MAD values, artificially lowering them.

 Reply: This is of course absolutely true. We indicate now that only for 14 samples anchoring was used. In addition, we include the citation of the data depository, in which it is possible to look up which samples this concern.

Line 264: Unless it is shown elsewhere, I would remove the suggestion that the magnetic record results from soil forming processes. Soil forming processes generate SP magnetite during pedogenesis. This would be reflected in IRM/X ratios or the ARM record, which are mentioned to be stable. I would simply state that the downcore records in X, ARM, and IRM reflect detrital input.

 Reply: Thank you for pointing this out. We changed the wording as suggested.

Line 348: I don't understand this sentence "the core parts were probably twisted against each other during transport along the layering". Also, if the original fabric is altered, how can only the declination be affected, but not the RPI?

 Reply: We are sorry, but we may have expressed ourselves incorrectly. We mean that the core parts are rotated against each other along the Z-axis. If they were twisted, deformation could occur. This is not the case. We have changed the wording to express ourselves correctly.

Line 398: Wouldn't sediment compaction and compression also result in inclination shallowing especially at high latitudes?

 Reply: Of course, sediment compaction may result in inclination shallowing. This is well known and we claim that it may be the cause for the shallower inclination at the side compared to GAD (lines 341-344). In turn, inclination shallowing may have an impact on RPI. We describe this issue in line 398-399, when we claim that a misalignment of magnetic minerals caused by compaction may be responsible for lower RPI in the lower part of the core.

Line 403: Awkward use of the term "final proof". I'd suggest another term like "additional evidence" or "further demonstrating"

 Reply: Thank you for pointing this out. We changed the wording into "contribute further evidence"

Line 532: I am curious as to why the authors suggest the change in TOC in Lake Levinson-Lessing is associated with the Bølling–Allerød interstadial in the NGRIP ice core rather than the transition out of the Younger Dryas into the Holocene? It does agree better with the C14 ages in table 1, but the authors have already stated that these could be too old. As the determination of the 18cm lock in

and the lock in discussion depends on this correlation, I'd like the authors to explain why the B/A and not the termination (which would presumably have a larger and lasting effect on TOC than the transient interstadial) was used as the tie point.

> **Reply:** The provided correlation scheme is reinforced by palynological evidence published in Lenz et al. 2022. Since the study is accepted and online available now (https://doi.org/10.1002/jqs.3384), we included the reference in the manuscript, and refer to this study for detailed explanations on this question.

We hope that we have now fully answered all open questions. We would very much appreciate it if this article could still be given a publication date in 2021 so that it is cited in existing works with the correct year.

Kind regards,
Stephanie Scheidt and Co-authors